# The Potential of Soluble Gas Stabilization (SGS) Technology in a Simulated Post-Frying Cooling Step of Commercial Fish Cakes

**DOI:** 10.3390/foods12142788

**Published:** 2023-07-22

**Authors:** Bjørn Tore Rotabakk, Elena Marie Rognstad, Anita Nordeng Jakobsen, Jørgen Lerfall

**Affiliations:** 1Nofima AS—Norwegian Institute of Food, Fisheries and Aquaculture Research, P.O. Box 8034, 4068 Stavanger, Norway; bjorn.tore.rotabakk@nofima.no; 2Department of Biotechnology and Food Science, NTNU—Norwegian University of Science and Technology, 7491 Trondheim, Norway; elenamarierognstad@outlook.com (E.M.R.); anita.n.jakobsen@ntnu.no (A.N.J.)

**Keywords:** modified atmosphere, soluble gas stabilization, minced fish, chilling

## Abstract

Soluble gas stabilization (SGS) technology is a novel way to increase the effectiveness of modified atmosphere (MA) packaging. However, SGS can be time-consuming and difficult to include in an existing process. This can be overcome by including CO_2_ in an existing processing step, such as the product’s cooling step. A full factorial design was set up with SGS times (0.5, 1.0, and 2.0 h) and temperatures of fish cakes (chilled (0 °C) or during chilling (starting at 85 °C)) as factors. MA-packaged fish cakes were included as a control. The response was headspace gas composition at equilibrium. Headspace gas composition at equilibrium showed significantly (*p* < 0.05) less dissolved CO_2_ in hot fish cakes after 0.5 h than in cold cakes. Still, no significant differences were found between hot and cold at 1.0 and 2.0 h. Also, all SGS samples, regardless of time and temperature, had a higher content of CO_2_ compared to modified atmosphere packaging (MAP).

## 1. Introduction

Soluble gas stabilization (SGS) technology, consisting of a pre-step before packing where carbon dioxide (CO_2_) is dissolved into the food matrix, has been demonstrated to be beneficial to prolonging the shelf life of muscle foods [1,2,3,4,5] and to improving the sustainability of modified atmosphere (MA) packaging due to a higher possible degree of filling (DF) [1,2]. However, appropriate well-designed industrial applications and commercial technological solutions still need to be made available. One concern the industry raises is related to changes in the processing logistics affecting the internal product flow, thereby reducing production efficiency. 

Diffusion is a time-consuming process affected by several factors, including the product shape [6], the headspace CO_2_ concentration [6,7], the chemical composition of the food matrix [8,9,10,11,12,13], the product and surrounding temperature [8,9], and the product salt content [7,14,15,16]. Depending on the mentioned factors, previous studies have shown that 3–4 days at refrigerated temperatures is necessary to obtain an equilibrium between the product headspace and the food matrix [8,13]. However, a positive effect on product quality or the potential to increase the DF can be obtained after shorter treatments, e.g., 1 to 2 h [12].

One possible way to implement SGS technology into a processing line for fish cakes without affecting production efficiency is to use the already-existing post-frying cooling step to dissolve CO_2_ into the food matrix. However, the practical challenge is the product’s initial temperature (85 °C) and the low CO_2_ solubility at such high temperatures. Due to fast cooling rates in the product surface and subsequently increased solubility during the first minutes of the cooling process, the potential to dissolve CO_2_ efficiently still exists. The present study aimed to study the possibility of implementing SGS technology in processing fish cakes using the post-frying cooling step to dissolve CO_2_ into the product before MA packaging.

## 2. Materials and Methods

The present study was carried out on commercial fish cakes purchased from a Norwegian supplier, and the experiment was performed the day after production. The fish cakes were not pre-treated by the supplier with CO_2_ or other packaging solutions that could affect the experimental results. The fish cakes consisted of silver smelt (*Argentina silus*) (51%) and haddock (*Melanogrammus aeglefinus*) (2%), in addition to milk (fat 4.5%), water, tapioca starch, rapeseed oil, onion, nutmeg extract, salt (1%), yeast extract, dextrose, and spices. The proximate composition as listed by the producer was fat (8%, of which 1.1% was saturated fatty acids); proteins (11%), carbohydrates (9.2%, of which 1.3% was sugar), and salt (1%), giving a total dry matter content of 29.2%.

The experimental setup (Figure 1) consisted of ‘initial temperature’ and ‘SGS processing time’ as experimental factors. The process started by heating the fried fish cakes in a steam cabinet (Convoterm, Elektrogeräte, Eglfing, Germany) to a core temperature of 85 °C. Twenty-four out of fifty-six samples were then treated via SGS immediately (96–97% CO_2_), simultaneously to them being cooled down for 0.5, 1.0, and 2.0 h, respectively (*n* = 8 for each group). 

The core temperature in four samples in each SGS treatment group was logged using a TrackSense Pro Double Sensor (Ellab AS, Hillerød, Denmark) during cooling. The last 32 samples were cooled down in plastic bags to 0 °C using wet ice. Then, 24 cold samples were treated with SGS according to their hot equivalents (0.5, 1.0, and 2.0 h, *n* = 8 for each group), however without dry CO_2_ as a cooling medium. The last eight samples were stored in air before packaging (0 h SGS time) and were included to simulate a standard commercial protocol and to work as a control group.

The SGS treatment was performed according to a modified method described by Rotabakk et al. [17]. Dry ice was used as the cooling medium for directly processing the heated fish cakes. The average headspace CO_2_ concentration during treatment was 99.3 ± 0.7%. After SGS treatment (Figure 1), all samples were re-packaged in MA-trays (C2187-F, Fearch Plastics, Holstebro, Denmark; OTR: 66–78 cm^3^ × 25 μm/m^2^/24 t/atm at 23 °C, and 0% relative humidity) using a Multivac T200 semiautomatic tray-sealing system (Sepp Haggenmüller SE & Co. KG, Wolfertschwenden, Germany) and a top film (Cryovac^®^ OSF33ZA, Sealed Air Food Care, Charlotte, NC, USA; OTR: 60 cm^3^/m^2^/24 t/bar at 23 °C, and 0% relative humidity) consisting of 99% polyethylene terephthalate (PET) and 1% polyethylene (PE) for improved sealing properties. The initial headspace atmosphere was adjusted to 60% CO_2_ (60.3 ± 0.3%) and balanced with N_2_. A total weight of 263 ± 2.6 g fish cake was packaged in each tray (4 pieces in each tray), giving a DF of 37.3 ± 0.3%. 

Both the SGS treatment and the storage (96 h) for the packaged samples took place in a chilled storage room with a temperature of 1.8 ± 0.3 °C.

The solubility of CO_2_ in fish cakes as affected by the experimental factors was measured as changes in headspace gas composition at the point of packaging (*n* = 17) and at equilibrium (*n* = 8 per group, 56 in total) by a Checkmate 9900 analyzer (PBI Dansensor, Ringsted, Denmark). The sample dry matter (DM) was measured according to ISO.6496 [18].

One-way ANOVA and Tukey’s comparison test were used to compare different groups. All statistical analyses were performed using Minitab 19 (Minitab Inc., State College, PA, USA). The alpha level was set to 5% (*p* < 0.05), and all results are given as an average ± standard deviation (SD), unless otherwise stated.

## 3. Results and Discussion

SGS treatment significantly (*p* < 0.001, one-way ANOVA) increased the level of CO_2_ in the tray headspace in all SGS samples compared to MAP (Figure 2). The solubility of CO_2_ in a food matrix is proportional to the concentration of CO_2_ in the headspace, which is determined by the solubility constant (Henry’s constant), as described by Henry’s Law [19]. However, this only applies when the system is in equilibrium. The measurements of headspace CO_2_ concentration were hence conducted after 96 h of storage, assuming that equilibrium was reached [8,12]. Furthermore, a significant increase in the headspace’s CO_2_ level (*p* < 0.001, one-way ANOVA) was observed as a function of a longer SGS processing time (0.5 h < 1.0 h < 2.0 h).

As the headspace CO_2_ increases, the amount of dissolved CO_2_ in the fish cakes increases accordingly, which implies that the fish cakes can dissolve CO_2_ during cooling. Also, there was a significantly (*p* < 0.001, one-way ANOVA) lower concentration of CO_2_ in fish cakes that were SGS-treated during chilling, compared to those treated hot at the processing time of 0.5 h, but not for those treated for 1.0 h or 2.0 h. Henry’s constant is dependent on several factors, amongst other temperatures [20], and the solubility of CO_2_ is known to increase with decreasing temperature. The average core temperature in the warm fish cake was 71.9 ± 2.3 °C at the starting point of the SGS treatment. The highest recorded core temperatures after 0.5, 1.0, and 2.0 h SGS treatment were 25.6 °C, 13.1 °C, and 5.0 °C, respectively (Figure 3). 

CO_2_ is a gas that has good solubility in both water and liquid fat in food [8,16]. Moreover, the solubility depends on the product’s total amount of water and fat. The dry matter in the fish cakes was 29.6 ± 0.1%, and no significant differences (*p* = 0.150, one-way ANOVA) were found between the cakes processed warm or cold. The fat content was approximately 8%, according to the producer. The primary ingredient in the fish cakes, silver smelt mince, consisted mainly of saturated and monounsaturated fatty acids in the range of C14 to C22 [8]. Since most of the present lipids will change their state during cooling, from a liquid form in warm fish cakes to a solid state after cooling, the improved solubility of CO_2_ is reasonable in the first part of the cooling procedure [8]. However, high temperatures naturally decrease the solubility of CO_2_ dramatically [9]. The higher temperature (25.6 °C) measured in the hot SGS-processed fish cakes (0.5 h group) was most likely the main reason for the observed difference between the pre-cooled samples and the samples cooled for 0.5 h, as the cakes’ proximate composition was equal. According to Fick’s second law of diffusion [21], CO_2_ starts to dissolve on the surface and then diffuses into the food matrix simultaneously, as the surface temperature will drop sooner than the fish cakes’ core temperature. The cooling of the fish cakes was slow enough in this system to show a significant difference after 0.5 h (showing the highest headspace CO_2_ concentration of cold-treated samples). However, this effect was insignificant after SGS processing times of 1.0 and 2.0 h. In a commercial setup, the cooling will take place in a much shorter timeframe. The current fish cake producer has a 40 min process directly after the frying step to cool the cakes down to refrigerated temperatures (<4 °C). It is likely that more efficient cooling for 40 min will result in an equal absorbance of CO_2_ as with the precooled fish cakes, as the surface and the first millimeters below will decrease in temperature more rapidly than observed in the experimental setup conducted in the present study. 

Increasing the headspaces’ CO_2_ level from 30.2% (MAP samples) to 34.8% with SGS during cooling (0.5 h SGS time) will most likely affect the product’s shelf life. It is well known that the effectiveness of MAP in inhibiting the growth of microorganisms is determined by the concentration of dissolved CO_2_ in the product [22]. Increasing the headspace concentration of CO_2_ from 38.8 to 44.9 significantly increased chicken breasts’ microbial and sensorial shelf life [17]. In addition, an increase from 51.5% to 56.8% in chicken drumsticks significantly reduced the amount of total aerobic bacteria and *Pseudomonas* spp. [23]. Both Rotabakk et al. [17] and Al-Nehlawi et al. [22] used an SGS processing time of 2.0 h, comparable with the design of the present experiment. It has also been previously shown that increased SGS time is followed by an increased amount of dissolved CO_2_. SGS times up to 48 h have been tested [24], showing that the dissolving rate of CO_2_ decreases with increased SGS processing time applied. In other words, by applying the SGS technology, the highest amount of CO_2_ will be dissolved at the beginning of the process.

This highlights the possibility of increasing the shelf life of fish cakes even after 0.5 or 1.0 h of SGS processing as part of the existing post-frying cooling step in the commercial production of fish cakes.

## 4. Conclusions

This trial demonstrates that it is possible to simultaneously chill down and dissolve CO_2_ into a newly fried product. It shows the possibility of including SGS technology in an already-existing production step without adding extra processing time. This is a big step toward an industrial application. Further work will be focused on scaling up and finding a secure way to maintain health and safety issues working with CO_2_ in a processing environment.

## Figures and Tables

**Figure 1 foods-12-02788-f001:**
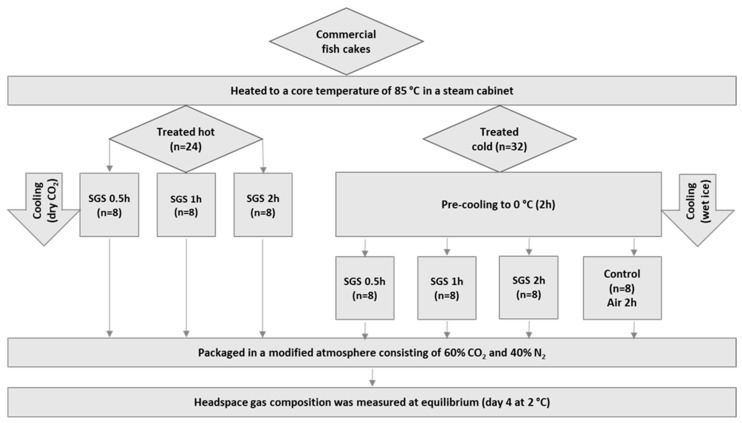
The experimental design. Commercial fish cakes were heated to 85 °C before being placed in 100% CO_2_ to cool down for 0.5, 1.0, and 2.0 h, respectively. Cold-processed samples (*n* = 32) were cooled to 0 °C before the soluble gas stabilization (SGS) treatment. Control samples were stored in air for 2 h before being packaged in a modified atmosphere (60% CO_2_:40% N_2_).

**Figure 2 foods-12-02788-f002:**
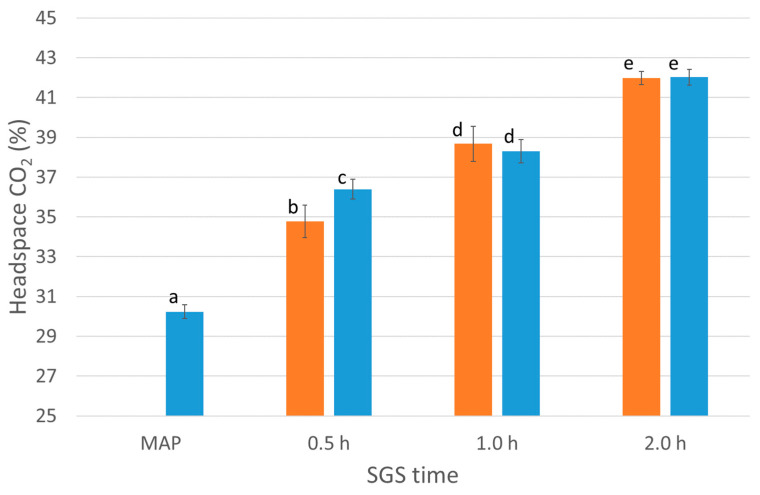
The headspace CO_2_ concentration of MAP (control) and SGS (0.5, 1.0, and 2.0 h) samples hot-treated (orange) or cold-treated (blue) after 96 h storage, presented as means ± SD. Columns with different lower superscript letters are significantly (*p* < 0.05) different as compared via one-way ANOVA and Tukey’s HSD test.

**Figure 3 foods-12-02788-f003:**
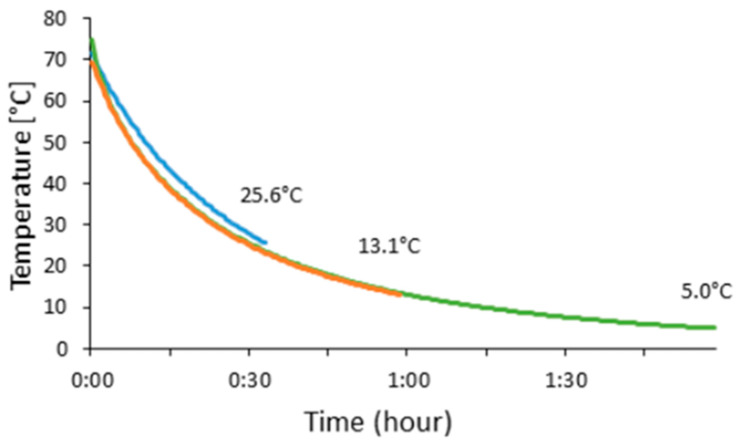
The core temperature of fish cakes (*n* = 8) was measured during SGS treatment in combination with cooling (0.5h SGS = blue, 1.0 h SGS = orange, and 2.0 h SGS = green).

## Data Availability

The data presented in this study are available upon request from the corresponding author.

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
