# Peer review of "The Potential of Soluble Gas Stabilization (SGS) Technology in a Simulated Post-Frying Cooling Step of Commercial Fish Cakes"

_foods, 2023, doi:10.3390/foods12142788_

Round 1

Reviewer 1 Report

The potential of soluble gas stabilization (SGS) technology in a simulated post frying cooling step of commercial fish cakes

The authors studied the possibility of implementing the modified Soluble Gas Stabilization atmosphere technology in processing of fish cakes using the post-frying cooling step to dissolve CO2 into the product before modified atmosphere packaging.

Comments:

 The subject of the manuscript is interesting; however, it seems that the authors should improve the organization of the manuscript, the representation form of their findings and also improve the discussion about the obtained results. The representation of the results is not in an appropriate form and the manuscript is not well written. The manuscript suffers from Lack of interpretations about the research outcome. The paper is also full of the ambiguous sentences and grammatical mistakes. Some examples are as follows:

 Page 2, line 68:

-         The last 32 samples were cooled down using wet ice in plastic bags to 0 °C before 24 were treated with SGS according to their hot equivalents (0.5, 1, and 2 h, n=8 for each group), without dry CO2 as a cooling media.

 Page 2, line 73:

-         The SGS treatment was performed according to a modified method described by 73 Rotabakk, et al. [17], and added dry ice as cooling media.

 Page 4, line 132:

-         The hottest hot fish cake had a core temperature of 26.3 °C after 0.5 h, compared to the cold ones at 1.8 ± 0.3 °C.

 Page 4, line 137:

-         The cooling of the fish cakes was slow enough in this system to give an effect after 0.5 h, but not after 1.0 and 2.0 138 h.

 Page 4, line 143:

-         Increasing the CO2 level in the headspace from 30.2 % to 34.8 % with SGS during cooling can eighter be used to increase the shelf life or the DF.

The potential of soluble gas stabilization (SGS) technology in a simulated post frying cooling step of commercial fish cakes

  The authors studied the possibility of implementing the modified Soluble Gas Stabilization atmosphere technology in processing of fish cakes using the post-frying cooling step to dissolve CO2 into the product before modified atmosphere packaging.

 Comments:

 The subject of the manuscript is interesting; however, it seems that the authors should improve the organization of the manuscript, the representation form of their findings and also improve the discussion about the obtained results. The representation of the results is not in an appropriate form and the manuscript is not well written. The manuscript suffers from Lack of interpretations about the research outcome. The paper is also full of the ambiguous sentences and grammatical mistakes. Some examples are as follows:

 Page 2, line 68:

-         The last 32 samples were cooled down using wet ice in plastic bags to 0 °C before 24 were treated with SGS according to their hot equivalents (0.5, 1, and 2 h, n=8 for each group), without dry CO2 as a cooling media.

 Page 2, line 73:

-         The SGS treatment was performed according to a modified method described by 73 Rotabakk, et al. [17], and added dry ice as cooling media.

 Page 4, line 132:

-         The hottest hot fish cake had a core temperature of 26.3 °C after 0.5 h, compared to the cold ones at 1.8 ± 0.3 °C.

 Page 4, line 137:

-         The cooling of the fish cakes was slow enough in this system to give an effect after 0.5 h, but not after 1.0 and 2.0 138 h.

 Page 4, line 143:

-         Increasing the CO2 level in the headspace from 30.2 % to 34.8 % with SGS during cooling can eighter be used to increase the shelf life or the DF.

   In my opinion, the manuscript (short communication) is not acceptable in its present form and the publication is not recommended.

 Author Response

We appreciate your valuable comments, and the revised manuscript has been reorganized, and typos and grammatical mistakes have been corrected to increase the readability of the manuscript. Moreover, additional content is also added to the discussion. See lines: 104-106, 112-116, 123-135, 140-143, 147-153, and 156-165.

Page 2, line 68:

  • The last 32 samples were cooled down using wet ice in plastic bags to 0 °C before 24 were treated with SGS according to their hot equivalents (0.5, 1, and 2 h, n=8 for each group), without dry CO2 as a cooling media.

Response: Many thanks for highlighting this example. The sentence has now been revised. See lines: 68-71

Page 2, line 73:

  • The SGS treatment was performed according to a modified method described by 73 Rotabakk, et al. [17], and added dry ice as cooling media.

Response: Many thanks for highlighting this example. The sentence has now been revised. See lines: 74-76.

Page 4, line 132:

  • The hottest hot fish cake had a core temperature of 26.3 °C after 0.5 h, compared to the cold ones at 1.8 ± 0.3 °C.

Response: Many thanks for highlighting this example. Due to the restructuration of the text, this sentence has been removed.

Page 4, line 137:

  • The cooling of the fish cakes was slow enough in this system to give an effect after 0.5 h, but not after 1.0 and 2.0 138 h.

Response: Many thanks for highlighting this example. The sentence has now been revised. See lines: 140-143

Page 4, line 143:

  • Increasing the CO2 level in the headspace from 30.2 % to 34.8 % with SGS during cooling can eighter be used to increase the shelf life or the DF.

Response: Many thanks for highlighting this example. The sentence has now been revised. See lines: 150-153

Reviewer 2 Report

This work is very innovative as it demonstrates that it is possible to simultaneously chill down and dissolve CO2 into a fried product. It shows that the SGS technology can be integrated into an existing production step without extending the processing time.

Minor suggestions:

Line 19 and Line 121 MAP - please write the meaning of this acronym

Line 77 – “66-78 cm3 x 25 μm/ m2/ 24 t/ atm at 23°C” please change to “66-78 cm3 x 25 μm/ m2/ 24 t/ atm at 23°C”

Line 80 “OTR: 60 cm3/m2/ 24 t/ bar at 23 °C” please change to “OTR: 60 cm3/m2/ 24 t/ bar at 23 °C”

Line 81: “PET” please write the meaning of this acronym

Line 80: “PE” please write the meaning of this acronym

Line 99: “P=0.150” please change to “p=0.150”

Line 119: “P<0.001” please change to “p<0.001”

Line 125: “P<0.05” please change to “p<0.05”

Line 129: “p<0.001” please change to “p<0.001”

Line 150: “p<0.001,” please change to “p<0.001”

Author Response

Many thanks for the positive comment. We agree that the presented results move the SGS technology closer to a commercial application of the concept.

Minor suggestions:

Line 19 and Line 121 MAP - please write the meaning of this acronym

Revision completed. Please see line: 19

Line 77 – “66-78 cm3 x 25 μm/ m2/ 24 t/ atm at 23°C” please change to “66-78 cm3 x 25 μm/ m2/ 24 t/ atm at 23°C”

Revision completed. Please see line: 78

Line 80 “OTR: 60 cm3/m2/ 24 t/ bar at 23 °C” please change to “OTR: 60 cm3/m2/ 24 t/ bar at 23 °C”

Revision completed. Please see line: 81

Line 81: “PET” please write the meaning of this acronym

Revision completed. Please see line: 82

Line 80: “PE” please write the meaning of this acronym

Revision completed. Please see line: 83

Line 99: “P=0.150” please change to “p=0.150”

Revision completed. Please see line: 127

Line 119: “P<0.001” please change to “p<0.001”

Revision completed. Please see line: 105

Line 125: “P<0.05” please change to “p<0.05”

Revision completed. Please see line: 110

Line 129: “p<0.001” please change to “p<0.001”

No changes done since it is already stated correctly in the manuscript

Line 150: “p<0.001,” please change to “p<0.001”

No changes done since it is already stated correctly in the manuscript

Reviewer 3 Report

I have read with interest communication „The potential of soluble gas stabilization (SGS) technology in a simulated post frying cooling step of commercial fish cakes”. The presented research results show significant application potential. Of particular importance is the fact that the SGS method used for hot-treated cakes gives comparable results to those for cold-treated cakes. This may allow the SGS method to be included in the normal technological process without the need to plan an additional stage after cooling the product. As a remark for the continuation of the research, I would recommend conducting headspace CO2 tests not only after 96 h, but after a much longer time close to the declared shelf-life of fish cakes, despite the provisions in lines 35-37. This confirms the fact of equilibrium between the product headspace and the food matrix.

Author Response

Many thanks for these valuable and nice comments. Due to the circumstances, it is difficult to make any changes in the design of the presented study. However, we will follow the headspace gas composition throughout the product’s shelf life for further studies. 

Round 2

Reviewer 1 Report

According to the improvements which have been accomplished by the authors, I think that the manuscript is ready for publication.